# Wasserstein Variational Inference

**Luca Ambrogioni**[*]
Radboud University
l.ambrogioni@donders.ru.nl

**Umut Güçlü**[*]
Radboud University
u.guclu@donders.ru.nl

**Yağmur Güçlütürk**
Radboud University
y.gucluturk@donders.ru.nl

**Max Hinne**
University of Amsterdam
m.hinne@uva.nl

**Eric Maris**
Radboud University
e.maris@donders.ru.nl

**Marcel A. J. van Gerven**
Radboud University
m.vangerven@donders.ru.nl

## Abstract

This paper introduces Wasserstein variational inference, a new form of approximate Bayesian inference based on optimal transport theory. Wasserstein variational inference uses a new family of divergences that includes both f-divergences and the Wasserstein distance as special cases. The gradients of the Wasserstein variational loss are obtained by backpropagating through the Sinkhorn iterations. This technique results in a very stable likelihood-free training method that can be used with implicit distributions and probabilistic programs. Using the Wasserstein variational inference framework, we introduce several new forms of autoencoders and test their robustness and performance against existing variational autoencoding techniques.

## 1 Introduction

Variational Bayesian inference is gaining a central role in machine learning. Modern stochastic variational techniques can be easily implemented using differentiable programming frameworks [1–3]. As a consequence, complex Bayesian inference is becoming almost as user friendly as deep learning [4, 5]. This is in sharp contrast with old-school variational methods that required model-specific mathematical derivations and imposed strong constraints on the possible family of models and variational distributions. Given the rapidness of this transition it is not surprising that modern variational inference research is still influenced by some legacy effects from the days when analytical tractability was the main concern. One of the most salient examples of this is the central role of the (reverse) KL divergence [6, 7]. While several other divergence measures have been suggested [8–12], the reverse KL divergence still dominates both research and applications. Recently, optimal transport divergences such as the Wasserstein distance [13, 14] have gained substantial popularity in the generative modeling literature as they can be shown to be well-behaved in several situations where the KL divergence is either infinite or undefined [15–18]. For example, the distribution of natural images is thought to span a sub-manifold of the original pixel space [15]. In these situations Wasserstein distances are considered to be particularly appropriate because they can be used for fitting degenerate distributions that cannot be expressed in terms of densities [15].

In this paper we introduce the use of optimal transport methods in variational Bayesian inference. To this end, we define the new *c-Wasserstein* family of divergences, which includes both Wasserstein

---

[*]These authors contributed equally to this paper.

metrics and all f-divergences (which have both forward and reverse KL) as special cases. Using this family of divergences we introduce the new framework of *Wasserstein variational inference*, which exploits the celebrated Sinkhorn iterations [19, 20] and automatic differentiation. Wasserstein variational inference provides a stable gradient-based black-box method for solving Bayesian inference problems even when the likelihood is intractable and the variational distribution is implicit [21, 22]. Importantly, as opposed to most other implicit variational inference methods [21–24], our approach does not rely on potentially unstable adversarial training [25].

## 1.1 Background on joint-contrastive variational inference

We start by briefly reviewing the framework of joint-contrastive variational inference [23, 21, 9]. For notational convenience we will express distributions in terms of their densities. Note however that those densities could be degenerate. For example, the density of a discrete distribution can be expressed in terms of delta functions. The posterior distribution of the latent variable $z$ given the observed data $x$ is $p(z|x) = p(z,x)/p(x)$. While the joint probability $p(z,x)$ is usually tractable, the evaluation of $p(x)$ often involves an intractable integral or summation. The central idea of variational Bayesian inference is to minimize a divergence functional between the intractable posterior $p(z|x)$ and a tractable parametrized family of variational distributions. This form of variational inference is sometimes referred to as posterior-contrastive. Conversely, in joint-contrastive inference the divergence to minimize is defined between two structured joint distributions. For example, using the reverse KL we have the following cost functional:

$$D_{KL}(p(x,z)\|q(x,z)) = \mathbb{E}_{q(x,z)}\left[\log \frac{q(x,z)}{p(x,z)}\right] , \qquad (1)$$

where $q(x,z) = q(z|x)k(x)$ is the product between the variational posterior and the sampling distribution of the data. Usually $k(x)$ is approximated as the re-sampling distribution of a finite training set, as in the case of variational autoencoders (VAE) [26]. The advantage of this joint-contrastive formulation is that it does not require the evaluation of the intractable distribution $p(z|x)$. Joint-contrastive variational inference can be seen as a generalization of amortized inference [21].

## 1.2 Background on optimal transport

Intuitively speaking, optimal transport divergences quantify the distance between two probability distributions as the cost of transporting probability mass from one to the other. Let $\Gamma[p,q]$ be the set of all bivariate probability measures on the product space $X \times X$ whose marginals are $p$ and $q$ respectively. An optimal transport divergence is defined by the following optimization:

$$W_c(p,q) = \inf_{\gamma \in \Gamma[p,q]} \int c(x_1,x_2) \, \mathrm{d}\gamma(x_1,x_2) , \qquad (2)$$

where $c(x_1,x_2)$ is the cost of transporting probability mass from $x_1$ to $x_2$. When the cost is a metric function the resulting divergence is a proper distance and it is usually referred to as the Wasserstein distance. We will denote the Wasserstein distance as $W(p,q)$.

The computation of the optimization problem in Eq. 2 suffers from a super-cubic complexity. Recent work showed that this complexity can be greatly reduced by adopting entropic regularization [20]. We begin by defining a new set of joint distributions:

$$U_\epsilon[p,q] = \left\{ \gamma \in \Gamma[p,q] \mid D_{KL}(\gamma(x,y)\|p(x)q(y)) \leq \epsilon^{-1} \right\} . \qquad (3)$$

These distributions are characterized by having the mutual information between the two variables bounded by the regularization parameter $\epsilon^{-1}$. Using this family of distributions we can define the entropy regularized optimal transport divergence:

$$W_{c,\epsilon}(p,q) = \inf_{u \in U_\epsilon[p,q]} \int c(x_1,x_2) \, \mathrm{d}u(x_1,x_2) . \qquad (4)$$

This regularization turns the optimal transport into a strictly convex problem. When $p$ and $q$ are discrete distributions the regularized optimal transport cost can be efficiently obtained using the Sinkhorn iterations [19, 20]. The $\epsilon$-regularized optimal transport divergence is then given by:

$$W_{c,\epsilon}(p,q) = \lim_{t \to \infty} \mathcal{S}_t^\epsilon[p,q,c] , \qquad (5)$$

where the function $\mathcal{S}_t^\epsilon[p,q,c]$ gives the output of the $t$-th Sinkhorn iteration. The pseudocode of the Sinkhorn iterations is given in Algorithm 1. Note that all the operations in this algorithm are differentiable.

**Algorithm 1** Sinkhorn Iterations. $C$: Cost matrix, $t$: Number of iterations, $\epsilon$: Regularization strength

---
1: **procedure** SINKHORN($C, t, \epsilon$)
2:     $K = \exp(-C/\epsilon),\ \ n, m = \text{shape}(C)$
3:     $r = \text{ones}(n, 1)/n,\ \ c = \text{ones}(m, 1)/m,\ \ u_0 = r,\ \ \tau = 0$
4:     **while** $\tau \leq t$ **do**
5:         $a = K^T u_\tau$                                              ▷ Juxtaposition denotes matrix product
6:         $b = c/a$                                                        ▷ "/" denotes entrywise division
7:         $u_{\tau+1} = m/(Kb),\ \ \tau = \tau + 1$
         $v = c/(K^T u_t),\ \ \mathcal{S}_t^\epsilon = \text{sum}(u_t * (K * C)v)$        ▷ "*" denotes entrywise product
8:     **return** $\mathcal{S}_t^\epsilon$

---

## 2   Wasserstein variational inference

We can now introduce the new framework of Wasserstein variational inference for general-purpose approximate Bayesian inference. We begin by introducing a new family of divergences that includes both optimal transport divergences and f-divergences as special cases. Subsequently, we develop a black-box and likelihood-free variational algorithm based on automatic differentiation through the Sinkhorn iterations.

### 2.1   c-Wasserstein divergences

Traditional divergence measures such as the KL divergence depend explicitly on the distributions $p$ and $q$. Conversely, optimal transport divergences depend on $p$ and $q$ only through the constraints of an optimization problem. We will now introduce the family of *c-Wasserstein divergences* that generalize both forms of dependencies. A c-Wasserstein divergence has the following form:

$$W_C(p, q) = \inf_{\gamma \in \Gamma[p,q]} \int C^{p,q}(x_1, x_2)\, \mathrm{d}\gamma(x_1, x_2)\,, \tag{6}$$

where the real-valued functional $C^{p,q}(x_1, x_2)$ depends both on the two scalars $x_1$ and $x_2$ and on the two distributions $p$ and $q$. Note that we are writing this dependency in terms of the densities only for notational convenience and that this dependency should be interpreted in terms of distributions. The cost functional $C^{p,q}(x_1, x_2)$ is assumed to respect the following requirements:

1. $C^{p,p}(x_1, x_2) \geq 0, \forall x_1, x_2 \in \text{supp}(p)$
2. $C^{p,p}(x, x) = 0, \forall x \in \text{supp}(p)$
3. $\mathbb{E}_\gamma[C^{p,q}(x_1, x_2)] \geq 0, \forall \gamma \in \Gamma[p, q]\,,$

where $\text{supp}(p)$ denotes the support of the distribution $p$. From these requirements we can derive the following theorem:

**Theorem 1.** *The functional $W_C(p, q)$ is a (pseudo-)divergence, meaning that $W_C(p, q) \geq 0$ for all $p$ and $q$ and $W_C(p, p) = 0$ for all $p$.*

*Proof.* From property 1 and property 2 it follows that, when $p$ is equal to $q$, $C^{p,p}(x_1, x_2)$ is a non-negative function of $x$ and $y$ that vanishes when $x = y$. In this case, the optimization in Eq. 6 is optimized by the diagonal transport $\gamma(x_1, x_2) = p(x_1)\delta(x_1 - x_2)$. In fact:

$$W_C(p, p) = \int C^{p,p}(x_1, x_2)p(x_1)\delta(x_1 - x_2)\, \mathrm{d}x_1\, \mathrm{d}x_2$$

$$= \int C^{p,p}(x_1, x_1)p(x_1)\, \mathrm{d}x_1 = 0\,. \tag{7}$$

This is a global minimum since property 3 implies that $W_C(p, q)$ is always non-negative.    $\square$

All optimal transport divergences are part of the c-Wasserstein family, where $C^{p,q}(x, y)$ reduces to a non-negative valued function $c(x_1, x_2)$ independent from $p$ and $q$.

Proving property 3 for an arbitrary cost functional can be a challenging task. The following theorem provides a criterion that is often easier to verify:

**Theorem 2.** *Let $f : \mathbb{R} \to \mathbb{R}$ be a convex function such that $f(1) = 0$. The cost functional $C^{p,q}(x,y) = f(g(x,y))$ respects property 3 when $\mathbb{E}_\gamma[g(x,y)] = 1$ for all $\gamma \in \Gamma[p,q]$.*

*Proof.* The result follows directly from Jensen's inequality. $\square$

## 2.2  Stochastic Wasserstein variational inference

We can now introduce the general framework of Wasserstein variational inference. The loss functional is a c-Wasserstein divergence between $p(x,z)$ and $q(x,z)$:

$$\mathcal{L}_C[p,q] = W_C(p(z,x), q(z,x)) = \inf_{\gamma \in \Gamma[p,q]} \int C^{p,q}(x_1, z_1; x_2, z_2) \, \mathrm{d}\gamma(x_1, z_1; x_1, z_1) \,. \quad (8)$$

From Theorem 1 it follows that this variational loss is always minimized when $p$ is equal to $q$. Note that we are allowing members of the c-Wasserstein divergence family to be pseudo-divergences, meaning that $\mathcal{L}_C[p,q]$ could be 0 even if $p \neq q$. It is sometimes convenient to work with pseudo-divergences when some features of the data are not deemed to be relevant.

We can now derive a black-box Monte Carlo estimate of the gradient of Eq. 8 that can be used together with gradient-based stochastic optimization methods [27]. A Monte Carlo estimator of Eq 8 can be obtained by computing the discrete c-Wasserstein divergence between two empirical distributions:

$$\mathcal{L}_C[p_n, q_n] = \inf_\gamma \sum_{j,k} C^{p,q}(x_1^{(j)}, z_1^{(j)}, x_2^{(k)}, z_2^{(k)}) \gamma(x_1^{(j)}, z_1^{(j)}, x_2^{(k)}, z_2^{(k)}) \,, \quad (9)$$

where $(x_1^{(j)}, z_1^{(j)})$ and $(x_2^{(k)}, z_2^{(k)})$ are sampled from $p(x,z)$ and $q(x,z)$ respectively. In the case of the Wasserstein distance, we can show that this estimator is asymptotically unbiased:

**Theorem 3.** *Let $W(p_n, q_n)$ be the Wasserstein distance between two empirical distributions $p_n$ and $q_n$. For $n$ tending to infinity, there is a positive number $s$ such that*

$$\mathbb{E}_{pq}[W(p_n, q_n)] \lesssim W(p,q) + n^{-1/s} \,. \quad (10)$$

*Proof.* Using the triangle inequality and the linearity of the expectation we obtain:

$$\mathbb{E}_{pq}[W(p_n, q_n)] \leq \mathbb{E}_p[W(p_n, p)] + W(p,q) + \mathbb{E}_q[W(q, q_n)] \,. \quad (11)$$

In [28] it was proven that for any distribution $u$:

$$\mathbb{E}_u[W(u_n, u)] \leq n^{-1/s_u} \,, \quad (12)$$

when $s_u$ is larger than the upper Wasserstein dimension (see definition 4 in [28]). The result follows with $s = \max(s_p, s_q)$. $\square$

Unfortunately the Monte Carlo estimator is biased for finite values of $n$. In order to eliminate the bias when $p$ is equal to $q$, we use the following modified loss:

$$\tilde{\mathcal{L}}_C[p_n, q_n] = \mathcal{L}_C[p_n, q_n] - (\mathcal{L}_C[p_n, p_n] + \mathcal{L}_C[q_n, q_n])/2 \,. \quad (13)$$

It is easy to see that the expectation of this new loss is zero when $p$ is equal to $q$. Furthermore:

$$\lim_{n \to \infty} \tilde{\mathcal{L}}_C[p_n, q_n] = \mathcal{L}_C[p,q] \,. \quad (14)$$

As we discussed in Section 1.2, the entropy-regularized version of the optimal transport cost in Eq. 9 can be approximated by truncating the Sinkhorn iterations. Importantly, the Sinkhorn iterations are differentiable and consequently we can compute the gradient of the loss using automatic differentiation [17]. The approximated gradient of the $\epsilon$-regularized loss can be written as

$$\nabla \mathcal{L}_C[p_n, q_n] = \nabla \mathcal{S}_t^\epsilon[p_n, q_n, C_{p,q}] \,, \quad (15)$$

where the function $\mathcal{S}_t^\epsilon[p_n, q_n, C_{p,q}]$ is the output of $t$ steps of the Sinkhorn algorithm with regularization $\epsilon$ and cost function $C_{p,q}$. Note that the cost is a functional of $p$ and $q$ and consequently the gradient contains the term $\nabla C_{p,q}$. Also note that this approximation converges to the real gradient of Eq. 8 for $n \to \infty$ and $\epsilon \to 0$ (however the Sinkhorn algorithm becomes unstable when $\epsilon \to 0$).

# 3 Examples of c-Wasserstein divergences

We will now introduce two classes of c-Wasserstein divergences that are suitable for deep Bayesian variational inference. Moreover, we will show that the KL divergence and all f-divergences are part of the c-Wasserstein family.

## 3.1 A metric divergence for latent spaces

In order to apply optimal transport divergences to a Bayesian variational problem we need to assign a metric, or more generally a transport cost, to the latent space of the Bayesian model. The geometry of the latent space should depend on the geometry of the observable space since differences in the latent space are only meaningful as far as they correspond to differences in the observables. The simplest way to assign a geometric transport cost to the latent space is to pull back a metric function from the observable space:

$$C_{PB}^p(z_1, z_2) = d_x(g_p(z_1), g_p(z_2)) , \qquad (16)$$

where $d_x(x_1, x_2)$ is a metric function in the observable space and $g_p(z)$ is a deterministic function that maps $z$ to the expected value of $p(x|z)$. In our notation the subscript $p$ in $g_p$ denotes the fact that the distribution $p(z|x)$ and the function $g_p$ depend on a common set of parameters which are optimized during variational inference. The resulting pullback cost function is a proper metric if $g_p$ is a diffeomorphism (i.e. a differentiable map with differentiable inverse) [29].

## 3.2 Autoencoder divergences

Another interesting special case of c-Wasserstein divergence can be obtained by considering the distribution of the residuals of an autoencoder. Consider the case where the expected value of $q(z|x)$ is given by the deterministic function $h_q(z)$. We can define the latent autoencoder cost functional as the transport cost between the latent residuals of the two models:

$$C_{LA}^q(x_1, z_1; x_2, z_2) = d(z_1 - h_q(x_1), z_2 - h_q(x_2)) , \qquad (17)$$

where $d$ is a distance function. It is easy to check that this cost functional defines a proper c-Wasserstein divergence since it is non-negative valued and it is equal to zero when $p$ is equal to $q$ and $x_1, z_1$ are equal to $x_2, z_2$. Similarly, we can define the observable autoencoder cost functional as follows:

$$C_{OA}^p(x_1, z_1; x_2, z_2) = d(x_1 - g_p(z_1), x_2 - g_p(z_2)) , \qquad (18)$$

where again $g_p(z)$ gives the expected value of the generator. In the case of a deterministic generator, this expression reduces to

$$C_{OA}^p(x_1, z_1; x_2, z_2) = d(0, x_2 - g_p(z_2)) . \qquad (19)$$

Note that the transport optimization is trivial in this special case since the cost does not depend on $x_1$ and $z_1$. In this case, the resulting divergence is just the average reconstruction error:

$$\inf_{\gamma \in \Gamma[p]} \int d(0, x_2 - g_p(z_2)) \, \mathrm{d}\gamma = \mathbb{E}_{q(x,z)}[d(0, x - g_p(z))] . \qquad (20)$$

As expected, this is a proper (pseudo-)divergence since it is non-negative valued and $x - g_p(z)$ is always equal to zero when $x$ and $z$ are sampled from $p(x, z)$.

## 3.3 f-divergences

We can now show that all f-divergences are part of the c-Wasserstein family. Consider the following cost functional:

$$C_f^{p,q}(x_1, x_2) = f\left(\frac{p(x_2)}{q(x_2)}\right) , \qquad (21)$$

where $f$ is a convex function such that $f(0) = 1$. From Theorem 2 it follows that this cost functional defines a valid c-Wasserstein divergence. We can now show that the c-Wasserstein divergence defined by this functional is the $f$-divergence defined by $f$. In fact

$$\inf_{\gamma_X \in \Gamma[p,q]} \int f\left(\frac{p(x_2)}{q(x_2)}\right) \mathrm{d}\gamma_X(x_1, x_2) = \mathbb{E}_{q(x_2)}\left[f\left(\frac{p(x_2)}{q(x_2)}\right)\right] , \qquad (22)$$

since $q(x_2)$ is the marginal of all $\gamma(x_1, x_2)$ in $\Gamma[p, q]$.

# 4   Wasserstein variational autoencoders

We will now use the concepts developed in the previous sections in order to define a new form of autoencoder. VAEs are generative deep amortized Bayesian models where the parameters of both the probabilistic model and the variational model are learned by minimizing a joint-contrastive divergence [26, 30, 31]. Let $\mathcal{D}_p$ and $\mathcal{D}_q$ be parametrized probability distributions and $\boldsymbol{g}_p(z)$ and $\boldsymbol{h}_q(x)$ be the outputs of deep networks determining the parameters of these distributions. The probabilistic model (decoder) of a VAE has the following form:

$$p(z, x) = \mathcal{D}_p(x | \boldsymbol{g}_p(z)) \, p(z) \,, \tag{23}$$

The variational model (encoder) is given by:

$$q(z, x) = \mathcal{D}_q(z | \boldsymbol{h}_q(x)) \, k(x) \,. \tag{24}$$

We can define a large family of objective functions of VAEs by combining the cost functionals defined in the previous section. The general form is given by the following total autoencoder cost functional:

$$
\begin{aligned}
C_{\boldsymbol{w},f}^{p,q}(x_1, z_1; x_2, z_2) = \; & w_1 d_x(x_1, x_2) + w_2 C_{PB}^p(z_1, z_2) + w_3 C_{LA}^p(x_1, z_1; x_2, z_2) \\
& + w_4 C_{OA}^q(x_1, z_1; x_2, z_2) + w_5 C_f^{p,q}(x_1, z_1; x_2, z_2) \,,
\end{aligned}
\tag{25}
$$

where $\boldsymbol{w}$ is a vector of non-negative valued weights, $d_x(x_1, x_2)$ is a metric on the observable space and $f$ is a convex function.

# 5   Connections with related methods

In the previous sections we showed that variational inference based on f-divergences is a special case of Wasserstein variational inference. We will discuss several theoretical links with some recent variational methods.

## 5.1   Operator variational inference

Wasserstein variational inference can be shown to be a special case of a generalized version of operator variational inference [10]. The (amortized) operator variational objective is defined as follows:

$$\mathcal{L}_{OP} = \sup_{f \in \mathfrak{F}} \zeta(\mathbb{E}_{q(x,z)}[\mathcal{O}^{p,q} f]) \tag{26}$$

where $\mathfrak{F}$ is a set of test functions and $\zeta(\cdot)$ is a positive valued function. The dual representation of the optimization problem in the c-Wasserstein loss (Eq. 6) is given by the following expression:

$$W_c(p, q) = \sup_{f \in L_C} \left[ \mathbb{E}_{p(x,z)}[f(x, z)] - \mathbb{E}_{q(x,z)}[f(x, z)] \right] \,, \tag{27}$$

where

$$L_C[p, q] = \{ f : X \to \mathbb{R} \,|\, f(x_1, z_1) - f(x_2, z_2) \leq C^{p,q}(x_1, z_1; x_2, z_2) \} \,. \tag{28}$$

Converting the expectation over $p$ to an expectation over $q$ using importance sampling, we obtain the following expression:

$$W_c(p, q) = \sup_{f \in L_C[p,q]} \left[ \mathbb{E}_{q(x,z)} \left[ \left( \frac{p(x, z)}{q(x, z)} - 1 \right) f(x, z) \right] \right] \,, \tag{29}$$

which has the same form as the operator variational loss in Eq. 26 with $t(x) = x$ and $\mathcal{O}^{p,q} = p/q - 1$. Note that the fact that $\zeta(\cdot)$ is not positive valued is irrelevant since the optimum of Eq. 27 is always non-negative. This is a generalized form of operator variational loss where the functional family can now depend on $p$ and $q$. In the case of optimal transport divergences, where $C^{p,q}(x_1, z_1; x_2, z_2) = c(x_1, z_1; x_2, z_2)$, the resulting loss is a special case of the regular operator variational loss.

## 5.2 Wasserstein autoencoders

The recently introduced Wasserstein autoencoder (WAE) uses a regularized optimal transport divergence between $p(x)$ and $k(x)$ in order to train a generative model [32]. The regularized loss has the following form:

$$\mathcal{L}_{WA} = \mathbb{E}_{q(x,z)}[c_x(x, g_p(z))] + \lambda D(p(z)\|q(z)) \ , \tag{30}$$

where $c_x$ does not depend on $p$ and $q$ and $D(p(z)\|q(z))$ is an arbitrary divergence. This loss was not derived from a variational Bayesian inference problem. Instead, the WAE loss is derived as a relaxation of an optimal transport loss between $p(x)$ and $k(x)$:

$$\mathcal{L}_{WA} \approx W_{c_x}(p(x), k(x)) \ . \tag{31}$$

When $D(p(z)\|q(z))$ is a c-Wasserstein divergence, we can show that the $\mathcal{L}_{WA}$ is a Wasserstein variational inference loss and consequently that Wasserstein autoencoders are approximate Bayesian methods. In fact:

$$\mathbb{E}_{q(x,z)}[c_x(x, g_p(x))] + \lambda W_{C_z}(p(z), q(z)) = \inf_{\gamma \in \Gamma[p,q]} \int [c_x(x_2, g_p(z_2)) + \lambda C_z^{p,q}(z_1, z_2)] \, d\gamma \ . \tag{32}$$

In the original paper the regularization term $D(p(z)\|q(z))$ is either the Jensen-Shannon divergence (optimized using adversarial training) or the maximum mean discrepancy (optimized using a reproducing kernel Hilbert space estimator). Our reformulation suggests another way of training the latent space using a metric optimal transport divergence and the Sinkhorn iterations.

## 6 Experimental evaluation

We will now demonstrate experimentally the effectiveness and robustness of Wasserstein variational inference. We focused our analysis on variational autoecoding problems on the MNIST dataset. We decided to use simple deep architectures and to avoid any form of structural and hyper-parameter optimization for three main reasons. First and foremost, our main aim is to show that Wasserstein variational inference works off-the-shelf without user tuning. Second, it allows us to run a large number of analyses and consequently to systematically investigate the performance of several variants of the Wasserstein autoencoder on several datasets. Finally, it minimizes the risk of inducing a bias that disfavors the baselines. In our first experiment, we assessed the performance of our Wasserstein variation autoencoder against VAE, ALI and WAE. We used the same neural architecture for all models. The generative models were parametrized by three-layered fully connected networks (100-300-500-1568) with Relu nonlinearities in the hidden layers. Similarly, the variational models were parametrized by three-layered ReLu networks (784-500-300-100). The cost functional of our Wasserstein variational autoencoder (see Eq. 25) had the weights $w_1$, $w_2$, $w_3$ and $w_4$ different from zero. Conversely, in this experiment $w_5$ was set to zero, meaning that we did not use a f-divergence component. We refer to this model as 1111. We trained 1111 using $t = 20$ Sinkhorn iterations. We evaluated three performance metrics: 1) mean squared reconstruction error in the latent space, 2) pixelwise mean squared reconstruction error in the image space and 3) sample quality estimated as the smallest Euclidean distance with an image in the validation set. Variational autoencoders are known to be sensitive to the fine tuning of the parameter regulating the relative contribution of the latent and the observable component of the loss. For each method, we optimized this parameter on the validation set. VAE, ALI and WAE losses have a single independent parameter $\alpha$: the relative contribution of the two components of the loss (VAE-loss/WAE-loss = $\alpha$*latent-loss + (1 - $\alpha$)*observable-loss, ALI-loss = $\alpha$*generator-loss + (1 - $\alpha$)*discriminator-loss for ALI). In the case of our 1111 model we reduced the optimization to a single parameter by giving equal weights to the two latent and the two observable losses (loss = $\alpha$*latent-loss + (1 -$\alpha$)*observable-loss). We estimated the errors of all methods with respect to all metrics with $alpha$ ranging from 0.1 to 0.9 in steps of 0.1. For each model we selected an optimal value of $\alpha$ by minimizing the sum of the three error metrics in the validation set (individually re-scaled by z-scoring). Fig. 1 shows the test set square errors both in the latent and in the observable space for the optimized models. Our model has better performance than both VAE and ALI with respect to all error metrics. WAE has lower observable error but higher sample error and slightly higher latent error. All differences are statistically significant (p<0.001, paired t-test).

In our second experiment we tested several other forms of Wasserstein variational autoencoders on three different datasets. We denote different versions of our autoencoder with a binary string

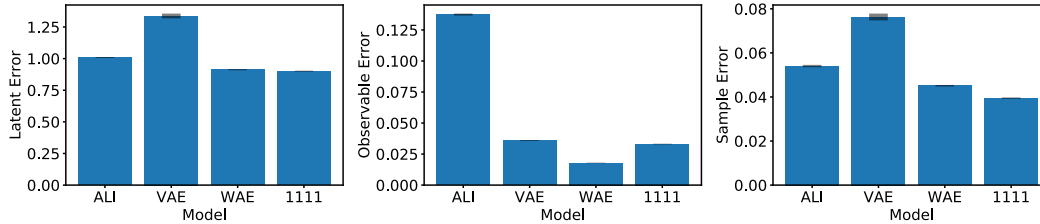

Figure 1: Comparison between Wasserstein variational inference, VAE, ALI and WAE.

Table 1: Detailed analysis on MNIST, fashion MNIST and Quick Sketches.

|  | MNIST | | | Fashion-MNIST | | | Quick Sketch | | |
|---|---|---|---|---|---|---|---|---|---|
|  | Latent | Observable | Sample | Latent | Observable | Sample | Latent | Observable | Sample |
| ALI | 1.0604 | 0.1419 | 0.0631 | 1.0179 | 0.1210 | 0.0564 | 1.0337 | 0.3477 | 0.1157 |
| VAE | 1.1807 | 0.0406 | 0.1766 | 1.7671 | 0.0214 | 0.0567 | 0.9445 | 0.0758 | 0.0687 |
| **1001** | 0.9256 | 0.0710 | 0.0448 | 0.9453 | 0.0687 | **0.0277** | 0.9777 | 0.1471 | **0.0654** |
| **0110** | 1.0052 | **0.0227** | 0.0513 | 1.4886 | 0.0244 | 0.0385 | 0.8894 | 0.0568 | 0.0743 |
| **0011** | 1.0030 | 0.0273 | 0.0740 | 1.0033 | **0.0196** | 0.0447 | 1.0016 | 0.0656 | 0.1204 |
| **1100** | 1.0145 | 0.0268 | 0.0483 | 1.3748 | 0.0246 | 0.0291 | 1.0364 | **0.0554** | 0.0736 |
| **1111** | 0.8991 | 0.0293 | **0.0441** | 0.9053 | 0.0258 | 0.0297 | **0.8822** | 0.0642 | 0.0699 |
| **h-ALI** | **0.8865** | 0.0289 | 0.0462 | **0.9026** | 0.0260 | 0.0300 | 0.8961 | 0.0674 | 0.0682 |
| h-VAE | 0.9007 | 0.0292 | 0.0442 | 0.9072 | 0.0227 | 0.0306 | 0.8983 | 0.0638 | 0.0677 |

denoting which weight was set to either zero or one. For example, we denote the purely metric version without autoencoder divergences as 1100. We also included two hybrid models obtained by combining our loss (1111) with the VAE and the ALI losses. These methods are special cases of Wasserstein variational autoencoders with non-zero $w_5$ weight and where the $f$ function is chosen to give either the reverse KL divergence or the Jensen-Shannon divergence respectively. Note that this fifth component of the loss was not obtained from the Sinkhorn iterations. As can be seen in Table 1, most versions of the Wasserstein variational autoencoder perform better than both VAE and ALI on all datasets. The 0011 has good reconstruction errors but significantly lower sample quality as it does not explicitly train the marginal distribution of $x$. Interestingly, the purely metric 1100 version has a small reconstruction error even if the cost functional is solely defined in terms of the marginals over $x$ and $z$. Also interestingly, the hybrid methods h-VAE and h-ALI have high performances. This result is promising as it suggests that the Sinkhorn loss can be used for stabilizing adversarial methods.

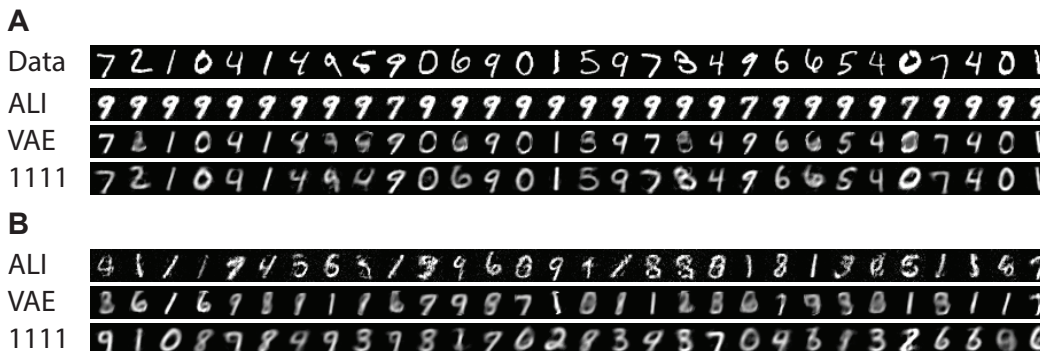

Figure 2: Observable reconstructions (A) and samples (B).

# 7 Conclusions

In this paper we showed that Wasserstein variational inference offers an effective and robust method for black-box (amortized) variational Bayesian inference. Importantly, Wasserstein variational inference is a likelihood-free method and can be used together with implicit variational distributions and differentiable variational programs [22, 21]. These features make Wasserstein variational inference particularly suitable for probabilistic programming, where the aim is to combine declarative general purpose programming and automatic probabilistic inference.

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
