[Reviews · NeurIPS 2018]

Reviewer 1



The authors describe a new framework for approximate inference called Wasserstain variational inference. This framework is based on the minimization of an entropic regularized version of the optimal transport cost through the Sinkhorn iterations. The result is a likelihood free training method which can be applied to implicit models. The proposed method is empirically evaluated on autoencoder models where it improves over two baselines: standard variational autoencoders and autoencoders trained with aversarially learned inference. Quality: The proposed method is theoretically well justified. However, I found that the main weakness is the experimental evaluation. The authors do not do hyper-parameter tuning of the different methods. Instead they just report results over an average across many different hyper-parameter values sampled from a prefixed sampling distribution. It is unclear wheather this non-standard evaluation was in detriment of the baselines. I believe a more riguorus evaluation protocol where hyper-parameters are optimized for each method (for example by grid search, or better by running a Bayesian optimization method) on some validation data is necessary. This can be expensive but it should be easy to paralellize over a computer cluster to reduce the extra cost. It seems that the baselines used by the authors are a bit weak, they just compare with standard VAEs (which are very old for the current pace of research in VAEs) and with a more recent technique which is not properly fine-tuned. I would recommend the authors to consier other alternatives. The performance metrics used by the authors to evaluate the quality of the different methods seem also a bit arbitrary. The authors could think of computing estimates of the marginal likelihood using importance weighted techniques. Also, how would their method work in the case of non-implicit models? Clarity: The paper is clearly written and easy to read. Novelty: The proposed approach seems novel and relevant. Significance: The theoretical contributions are significant. However, the empirical evaluation of the proposed method is too weak to clearly evaluate the significance of the proposed method. See my comments above. Update after rebuttal: After looking at the new experiments included in the rebuttal, where the authors perform hyper-parameter tuning and compare with additional baselines, I am supportive of acceptance.

Reviewer 2



MAIN IDEAS The paper introduces a new family of divergence measures, which it calls the c-Wasserstein family of divergences. In application (in this paper) the distance between a generative joint density p(x, z) and its approximation q(x, z) is minimized, where (x, z) ~ training data. The "transport cost" function of the c-Wasserstein divergence depends on distributions p and q, as well as its usual arguments. Theoretical properties are provided (all of this is novel work), and a Monte Carlo estimate shown. The MC estimate is biased for a finite sample size, and the paper shows how to remove the bias in one specific case when p = q. The gradients of this loss are obtainable by back-propagating through the so-called "Sinkhorn iterations" from optimal transport theory. RELATIONSHIP TO PAST WORK The paper builds nicely on Operator Variational Inference (from NIPS last year or the year before), and Renyi divergence variational inference (also at NIPS). Note to authors: the paper is very heavily biased toward citing work from one inner circle: 7/10 of the first cited papers are from the same group (!). Surely the world is larger? STRENGTHS - A beautiful new theoretical umbrella that ties f-divergences and Wasserstein divergences together. - Very well written paper. WEAKNESSES - In time gone by, or "the days still influenced by legacy effects", as the paper puts it in its opening paragraph, divergence measures in approximate inference would always be used with respect to a true partition function, exact moments, and so on. One would compare against a ground truth MCMC estimate, do some thermodynamic integration, etc. Now, the evaluation is very fuzzy: the authors look at the mean squared deviation in the latent space over 30 re-runs of each method. (We assume by mean squared deviation that 30 latent samples were drawn for each datapoint, for each method, and the deviation from the mean of those samples reported?) The paper could state why these metrics are insightful? A method that encodes x into a delta function at zero, would have a zero latent error, and so why is "outperforms" (Line 210) such a big deal? - The flexibility through C^{pq} introduces another layer of complexity that a modeller has to mitigate. - Is a bias introduced by running the Sinkhorn iterations for a finite number of steps? - The paper is very terse (especially section 5.1) and Typos: Line 7 in Algorithm 1, sum(u_\tau ... Line 139, should it not map to the expected value of p(x | z)? Line 181, equation, end, D(p(z) || q(z)) Line 224, J_e_nsen-Shannon

Reviewer 3



The paper presents a variant of Wasserstein distance between two probability distributions p and q called c-Wasserstein divergence which is defined with a functional C^{p,q} of p and q. The idea of incorporating two distributions into the cost function looks exciting. However, it is not clear how the newly defined divergence is related to Wasserstein distance. There is no analysis between these two divergences such as inequality, bound, or asymptotics. For example, Theorem 3 presents the property of Wasserstein estimator but what is the implication of this Theorem for c-Wasserstein divergence. The experimental section does not prove any benefit of c-Wasserstein divergence in compared with Wasserstein distance used with VAE as in [31].